# The [DPPH●/DPPH-H]-HPLC-DAD Method on Tracking the Antioxidant Activity of Pure Antioxidants and Goutweed (*Aegopodium podagraria* L.) Hydroalcoholic Extracts

**DOI:** 10.3390/molecules25246005

**Published:** 2020-12-18

**Authors:** Jolanta Flieger, Michał Flieger

**Affiliations:** 1Department of Analytical Chemistry, Medical University of Lublin, Chodźki 4a (Collegium Pharmaceuticum), 20-093 Lublin, Poland; 2Faculty of Medicine, Medical University of Lublin, Aleje Racławickie 1, 20-059 Lublin, Poland; michalflieeeger@gmail.com

**Keywords:** *Aegopodium podagraria* L., goutweed, antioxidant activity, DPPH, RP-HPLC

## Abstract

The 2,2-diphenyl-1-picrylhydrazyl (DPPH)-reverse phase (RP)-HPLC-diode array detector (DAD) method was tested on standard antioxidants (AOs), i.e., reduced glutathione (GSH), ascorbic acid (vitamin C), and alcoholic extracts of *A. podagraria* L. An elaborated HPLC procedure enabled the simultaneous measurement of the redox couple DPPH-R (2,2-diphenyl-1-picrylhydrazyl radical)/DPPH-H (2,2-diphenyl-1-picrylhydrazine). Both forms were fully separated (R_s_ = 2.30, *α* = 1.65) on a Zorbax Eclipse XDB-C18 column eluted with methanol–water (80:20, *v*/*v*) and detected at different wavelengths in the range of 200–600 nm. The absorbance increases of the DPPH-H as well as the DPPH-R peak inhibition were measured at different wavelengths in visible and UV ranges. The chromatographic method was optimized, according to reaction time (slow, fast kinetics), the linearity range of DPPH radical depending on the detection conditions as well as the kind of the investigated antioxidants (reference chemicals and the ground elder prepared from fresh and dry plants). The scavenging capacity was expressed by the use of percentage of peak inhibition and the IC_50_ parameters. The evaluated extracts displayed antioxidant activity, higher than 20% inhibition against 350 µM DPPH free radical. The results show that extract prepared from dry plants in the ultrasonic bath exhibits the highest antioxidant potential (IC_50_ = 64.74 ± 0.22 µL/mL).

## 1. Introduction

The role of oxidative stress as one of the key mechanisms of aging as well as in the pathogenesis of many diseases such as atherosclerosis, cancer, Alzheimer’s disease, and cardiovascular disorders has been confirmed [1,2,3,4,5]. It has been proven that oxidative stress appears as a consequence of the imbalance between the generation of reactive oxygen species (ROS) such as O_2_, H_2_O_2_, and OH, alongside reactive nitrogen species (RNS), and the antioxidant defenses. When endogenous antioxidants are involved in the defense mechanisms against the free radical, covering such compounds as uric acid, albumin, and metallothioneins or enzymes (superoxide dismutase, catalase) cannot ensure the protection of the organism against ROS, and the need arises for exogenous antioxidants.

Exogenous antioxidants derived from natural sources such as plants (flavonoids, polyphenols, carotenoids, vitamins) or minerals (selenium, zinc, manganese) can remove free radicals, inhibiting the adverse effects causing by ROS in the form of oxidation of lipids, protein, and nucleotides. In this way, they prevent damage done to the cells by ROS. Considering the beneficial role of the free radical-scavenging bioactive substances in human health protection, they have been still searched and constantly investigated. 

Many chemical assays have been described to evaluate the antioxidant capacities of either different chemicals or products of natural origin. These tests utilize, among others, 2,2-azinobis(3-ethyl-benzothiazoline-6-sulfonic acid) (ABTS), ferric reducing antioxidant power (FRAP), electron Spin Resonance (ESR), and oxygen radical absorption capacity (ORAC) [6,7,8,9,10,11,12].

The DPPH (2,2-diphenyl-1-picrylhydrazyl radical (DPPH•, DPPH-R) discovered by Goldsmith and Renn in 1922 [13], due to its stability and a high enough redox potential to oxidize the most common natural antioxidants, has been proposed for the assay of the antioxidant capacity of biological materials [14]. Since that time, a convenient spectrophotometric method with DPPH has been extensively reported to estimate the antioxidant capacity of chemicals as well as many products of natural origin [15,16,17,18,19,20]. The DPPH radical is reduced to 2,2-diphenyl-1-picrylhydrazine (DPPH-H) via attaching a hydrogen atom or an electron to the radical center owing to the reaction with an antioxidant. The antioxidants can react with DPPH• by different possible mechanisms such as HAT (hydrogen atom transfer) and multistep reactions such as SET (single electron transfer) and SPLET (sequential proton loss electron transfer) [19,20,21,22,23,24,25,26] or mixed [27] mechanisms. It was established that the SET and SPLET mechanisms are predominant in a non-aqueous environment mainly because the capacity of the organic solvent of creating hydrogen bonds with the antioxidants (AOs) [21,22,23,24,25,26,27,28,29]. More recently, Xie and Schaich [30] re-evaluated the DPPH spectrophotometric assay for screening the antioxidant activity of a series of antioxidants and natural extracts in different solvents and pH values.

Characteristic of the reaction is the color change from purple (DPPH• or DPPH-R) to yellow (DPPH-H). The spectrophotometric measurement of the disappearance of the violet color is made at the wavelength of *λ* = 517 nm. The measure of antioxidant power is the absorbance difference of the DPPH radical solution before and after the reaction with the antioxidant. The biggest disadvantage and limitation of the spectrophotometric method is the inability to measure colored samples. In order to avoid measurement interferences related to the analysis of colored samples, in 1998, Yamaguchi et al. [31] established a DPPH-HPLC method for determining the free radical-scavenging activity. The authors evaluated the free radical-scavenging activity of selected commercial beverages such as wines, different kinds of tea, and coffee by measuring the decrease of the DPPH peak detected at 517 nm. The authors expressed the radical-scavenging activity as the Trolox equivalent. The results of the concentration for 50% radical-scavenging activity of reference antioxidants measured by the DPPH-HPLC method were in very good agreement with those measured colorimetrically. In 2012 Boudier et al. [32] continued a DPPH radical reduction method combined with high-performance liquid reversed-phase chromatography (DPPH-HPLC). The DPPH-R/DPPH-H couple was separated on a C18 column and acetonitrile/10 mM ammonium citrate pH = 6.8 (70:30, *v*/*v*) mobile phase. In contrast to the previous study performed by Yamaguchi et al. [31], the detection was set within the UV range at 330 nm. Trolox (6-hydroxy-2,5,7,8-tetramethylchroman-2-carboxylic acid), ascorbic acid, and GSH were used as model antioxidants. For the first time, the measure of antioxidant capacity was the increase in the peak area of the DPPH reduced form before and after the reaction with the antioxidant. The evident advantage of the HPLC detection of the radical is the possibility to suppress interferences linked to colors of natural extracts. The elaborated method appears to be simple and without doubt superior to colorimetry due to non-interference from other pigments, but surprisingly, there are not many imitators and full studies defining the measurement conditions [33,34]. Unfortunately, the stability of the DPPH-HPLC test, the precision of the method, and the linearity were considered to be the main problems of the evaluation of antioxidant properties by the HPLC method.

More and more researchers are drawing attention to the need to standardize the methods of determining the antioxidant potential, due to the fact that there are large discrepancies in the results obtained in different laboratories and the inability to compare different methods even for reference antioxidants. Mishra et al. [15] in the critical review showed that according to literature data, the IC_50_ of ascorbic acid ranges from 11.85 to 629 µM. The authors report that one of the reasons for these discrepancies is the arbitrary fixing of the reaction time to 20–30 min, while it may require longer to reach equilibrium. Furthermore, there is no consistency according to the concentration of DPPH, sample volume, AO concentration, measured parameters, and their units. The HPLC-DPPH assay procedure should be elaborated to properly design new investigations to evaluate the antiradical properties of plant extracts and appropriate reference standards. In the present study, the HPLC method with a DPPH-R/DPPH-H couple has been tested for evaluation of the free radical-scavenging activity of ascorbic acid or reduced glutathione as standard antioxidants (AOs) and *A. podagraria* extracts. 

Goutweed (*Aegopodium podagraria* L.) is a perennial plant of *Apiaceae* family, which grows in Europe, Asia, and North America. Due to its hypouricemic and uricosuric action as well as the one suppressing inflammation, goutweed was established as a principal herbal drug to treat gout, inflammatory states in kidneys and bladder, and to facilitate wound healing in traditional folk medicine [34].

Preparations obtained from the aerial part of goutweed as well as raw material are rich in bioactive secondary metabolites such as aliphatic C17-polyacetylenes including falcarinol and falcarindiol, which are responsible for anti-inflammatory and antimicrobial properties [35,36,37]. Their selective cytotoxic activity against cancer cells has also been confirmed in an in vivo model [38]. Among polyacetylenes of the falcarinol-type, which are the most numerous group of chemical compounds of the Apiaceae family’ plants, goutweed is also an appreciated source of hydroxycinnamic acids (caffeic, chlorogenic), essential oil components (mono- and sesquiterpenes mainly α-, β-pinene, and sabinene) [39,40], and flavonoids, coumarins, and carotenoids [41,42,43]. Furthermore, goutweed contains vitamins (ascorbic acid, tocopherols) and macro- and microelements, including zinc, iron, copper, chromium, manganese, cobalt, calcium, potassium, fluorine, and carbohydrates (glucose, fructose), lectins, and glycoproteins-binding carbohydrates [44,45]. However, the content of bioactive compounds in the raw material affecting the pharmacological activity depends on the vegetation period, the morphological part of the plant, and primarily on its origin. 

*A. podagraria* has still been extensively screened not only for its chemical content or pharmacological activity but also as an antioxidant active ingredient [42,46,47]. Recently, the effect of different extraction methods (ultrasonic-assisted extractions, extraction in a Soxhlet apparatus, extraction at the boiling point of the solvent), as well as extractive conditions (kind of solvent, extraction time, parts of the plant) on the antioxidative properties of the *A.podagraria* extracts has been reported by Wróblewska et al. [48]. 

The aim of this work was to perform standardization of the DPPH-RP-HPLC method considering the detection wavelength, the reaction time, and the concentration of DPPH for evaluation of the free radical-scavenging activity of two chosen reference chemicals such as ascorbic acid and glutathione reduced and ethanol *A. podagraria* extracts (leaves/stems; fresh; and air-dried). For this purpose, an HPLC with isocratic elution mode using a DPPH-R/DPPH-H couple has been tested.

## 2. Results and Discussion

### 2.1. HPLC-DAD Detection Conditions

The DPPH radical reduction test by high-performance liquid chromatography (DPPH-HPLC) was performed in a reversed phase system on a Zorbax Extend C18 column using 80% methanol. A methanolic solution of 0.7 mM DPPH-R was analyzed at different wavelengths (Figure 1). It turned out that the tested solution of DPPH free radical (DPPH-R) contains also a low concentration of its reduced form (DPPH-H). This may be due to the reaction of DPPH-R with the stationary phase or the metal parts of HPLC instruments. However, the exact cause of this phenomenon is not clearly explained. The presence of the above- mentioned contamination may be also related to impurities in the standard. Therefore, the DPPH-H peak surface appearing in the standard was subtracted in all further calculations.

The chromatographic system used ensures the retention of both forms of DPPH• (DPPH-R) and DPPH-H at a satisfactory level of separation: R_s_ = 2.30 (R_s_ > 1) and *α* = 1.65 (*α* > 1). As indicated in Figure 1, the diode array detector (DAD) allows the simultaneous collection of chromatograms at different wavelengths at a single run. The UV-Vis spectrum is in the 200–600 nm range of each separated peak: the free radical form (DPPH•) as well as its reduction product (DPPH-H) (Figure 2) is useful for selecting an optimal wavelength for the final HPLC method. Overlapped spectra intersect at 517 nm regardless of the DPPH-R/DPPH-H ratio, because the sum of DPPH-R + DPPH-H remains constant. At an isosbestic point, both forms of the compound in solution have equal molar absorption coefficients, which is why there are minimal disproportions between the sizes of the peaks. A wavelength of 517 nm was commonly selected for the spectrophotometric measurement of antioxidant properties. The reproducibility of the DPPH-HPLC method was checked by dosing 0.7 mM methanolic DPPH several times (*n* = 3). Quantitative determination of both forms of DPPH in a single run was based on the peak area measured at 517 nm. High precision was achieved at the RSD level: 2.28% and 3.41% for DPPH-H and DPPH-R, respectively. However, a diode array detector (DAD) permits the measurement of the reagent (DPPH•) and the product (DPPH-H) at their analytical wavelengths on the same chromatogram without affecting selectivity. That is why it is also possible to shift detection from the visible to UV range and measure DPPH-R at 330 nm and DPPH-H at 350 nm.

The range of linearity between the peak area and DPPH concentration is illustrated in Figure 3. The limits of quantification (LOQ) and detection (LOD) were calculated by the relationship between the standard deviation (*σ*) of the calibration curve and its slope (*S*) as suggested by the validation guideline ICH Q2(R1). The LOD and LOQ were calculated from the following equations: LOD = (3.3 × *σ/S*) and LOD = (10 × *σ/S*), where the standard deviation of the response (*σ*) can be determined based on the standard deviation of y-intercepts of regression lines, whereas *S* represents the slope of the calibration curve. A lowest LOD was obtained for DPPH-R (25.55 µM) at 330 nm. The linear regression parameters obtained for a DPPH-R/DPPH-H couple at different wavelengths were collected in Table 1.

The presence of the antioxidant enables us to determine the LOD value of DPPH-H form from the calibration curve. Figure 4 presents changes in the surface areas of the DPPH-R and DPPH-H peaks recorded at different wavelengths, after mixing DPPH with an excess of ascorbic acid. Obtained values are much lower in comparison to the LOD values measured for DPPH standard without any additives (Table 1). It can be concluded that measuring the quenching of the DPPH-R peak is much more sensitive in the presence of AO. 

The influence of the concentration of DPPH-R on the symmetry and efficiency of the chromatographic system may be analyzed on the basis of the data included in Table 2. As can be seen, the concentration of DPPH-R influences the retention only for very dilute solutions with concentrations lower than 200 µM. Further concentration increase neither influences the retention nor worsens the symmetricity of the peaks (0.8 < *A*s < 1.2) and system efficiency (*N* > 2000). However, the linearity range of the relationship between the peak area of the DPPH radical and its concentration varies at different wavelengths. The broadest range that ensures that the peak area is proportionate vs. concentration of DPPH-R shows the measurement taken at 517 nm.

### 2.2. Antioxidant Properties of Standard AOs Measured by the DPPH-HPLC-DAD

#### 2.2.1. Optimization of the Incubation Time

In order to test the DPPH-HPLC-DAD method, the antioxidant activities of ascorbic acid and reduced glutathione (GSH) as standard AOs were evaluated. The variation of scavenging activity with time is presented in Figure 4. Both AOs showed time-dependent DPPH radical-scavenging activities. It was generally observed that the DPPH radical-scavenging effect increased with time. However, whereas ascorbic acid required less than 15 min, the percent of inhibition measured for reduced glutathione toward the DPPH radical peak stabilized for longer than 60 min. Therefore, these antioxidants can be classified as fast (<30 min) and slow (>1 h) kinetics, respectively, according to the time duration of reaction to reach the steady state. As can be seen, transferring the detection into the ultra-violet range appears to be unbeneficial either for the sensitivity of the DPPH-R peak inhibition or DPPH-H peak increases under influence of antioxidants. It should be emphasized that the most sensitive detection wavelength to measure free radical scavenging expressed as the percentage of DPPH-R/DPPH-R’ peak area inhibition was in the visible region of the spectrum, namely 517 nm. Measuring the concentration for 50% radical-scavenging activity at a reasonable period of time (<1 h) is possible only for ascorbic acid, which is superior to glutathione reduced, representing slow reaction kinetics. IC_50_ values for ascorbic acid against 350 µM DPPH were calculated as 80.48 µg/mL for 330 nm and almost a six times higher value of 20.18 µg/mL for 517 nm corresponding to 457.0 µM (22.37% inhibition) and 114.35 µM (89.39%), respectively. The values reported by various studies, and collected by Mishra et al. [15], were in the range from 11.85 to 629 µM.

#### 2.2.2. Optimization of the AO Concentration

The radical-scavenging effect should have been constantly increased alongside with the AO concentration, which is visible as the inhibition of the peak area in the case of DPPH-R or excitation characteristic of the DPPH-H’ peak. This happens because after adding ascorbic acid acting as an electron donor, it consumes the free radical DPPH-R, reconstructing DPPH-H. Chromatograms of DPPH methanolic standard solution and products after reaction with ascorbic acid are shown in Figure 5.

When investigating the validation parameters of the DPPH-RP-HPLC method, the problem concerning the linearity range of calibration curves built for a quench of DPPH-R or increase of DPPH-H peak under the influence of the tested AOs appeared. As it can be seen from Figure 5, ascorbic acid addition gave rise to the same product as that previously detected, DPPH-H. Investigating dependencies of the AO concentration on the peak area both DPPH-R and DPPH-H (Figure 6), the stoichiometric ratio of the reaction of DPPH with ascorbic acid or reduced glutathione as 1:2, and 1:1 respectively can be confirmed [28,29]. The proportional consumption of DPPH-R was observed only when ascorbic acid concentration was higher than the DPPH initial concentration. In the case of the lower value of the AO concentration, the peak area of DPPH-R was surprisingly increasing. The probable reason for this increase is the SET or SPLET multistep mechanism of the reaction of DPPH with AO and the creation of intermediate products with the same retention [21,22,23,24,25,26,27,28,29,30]:SET: DPPH● + AH → DPPH^−^ + AH^+^●; AH^+^● → AH● +H^+^; DPPH^−^ + H^+^ → DPPH−H(1)
SPLET: AH → AH^−^ + H^+^; AH^−^ + DPPH● → AH● + DPPH^−^; DPPH^−^ + H^+^ → DPPH−H(2)

It has been already proven that the SET and SPLET mechanisms dominate in non-aqueous solutions because of the ability of organic solvents to form strong hydrogen bonds with AOs [29,32]. In turn, in aqueous solution, AO is able to quench the DPPH radical by hydrogen atom transfer (HAT):HAT: DPPH● + A → DPPH-H + A●(3)

Antioxidant capacity can be measured as the difference in the area of the peak of the radical form before and after the reaction or as an increase in the DPPH-H peak area. At the low antioxidant capacity of the investigated sample, it seems more advantageous to measure the inhibition of the DPPH-R peak than the increase in the DPPH-H peak, which remains constant above a certain concentration. In addition, there are four important considerations. (i) The DPPH-R standard is contaminated with a small amount of DPPH-H. (ii) To determine the antioxidant capacity by the increase of the DPPH-H peak, two standards are required: DPPH-R to quantify the reaction substrate and DPPH-H to quantify reaction product. (iii) To measure the increase of the DPPH-H peak, the area of the trace DPPH peak would have to be subtracted with each measurement. (iv) Since there is a lower LOD for DPPH-R (9.5 µM) in comparison to DPPH-H (19.2 µM), it can be concluded that measuring the quenching of the DPPH-R peak by AO appears to be more favorable in comparison to measurement based on the increase of the DPPH-H peak. Linear regression parameters for the detection conditions of AOs by DPPH-R quench at 517 nm are collected in Table 3.

### 2.3. The Determination of the Antioxidant Capacity of Aegopodium podagraria L. Extracts by DPPH-HPLC-DAD

So far, anti-inflammatory, antimicrobial, nephro- and hepatoprotective properties and potential anti-cancerogenic activities of goutweed extracts have been confirmed by independent research groups [49,50]. The beneficial effects of goutweed extracts on purine metabolism reflected by antihyperglycemic lipid-lowering have also been proven [50,51,52,53,54,55,56,57,58]. The obtained results testify to the efficacy of the goutweed in the treatment of diseases related to carbohydrate metabolism disorders, especially Type 2 diabetes [54]. In addition, the synergism of goutweed extract with the anti-diabetic drug metformin in dexamethasone-treated rats has been demonstrated. Tovchiga et al. [55,58] noticed an interesting link between the antihyperuricemic and psychotropic activity of goutweed preparations. They studied the sedative effects of goutweed extracts on the depression, anxiety, locomotor activity, exploratory behavior, and memory in mouse models. The results of the study demonstrated that *A. podagraria* extract at a dose of no higher than 100 mg/kg is potent enough to decrease the level of depression and anxiety in animals of both sexes [58]. 

As evidenced by previous studies, the extraction conditions determine the chemical composition of extracts [48,59] which, therefore, may show different pharmacological activity and may differ in the total antioxidant potential.

It is known that goutweed is a rich source of many biologically active substances such as polyacetylenes, terpenes, coumarins, polyphenolic compounds, micro and macro elements, and vitamins (ascorbic acid, α-tocopherol). It was found that the aerial parts of this plant have a significant lipophilic antioxidant content (146.07 m/100 g^−1^ fresh weight) similar to spinach and carotenes slightly lower than in nettle [48,60].

The evaluation of the antioxidant capacity of various goutweed extracts and the creation of their specific, bioactive antioxidant profile can help to select the most favorable conditions ensuring the extraction of natural antioxidants with pro-health effects.

To prepare the extracts, the maceration of ground fresh plant material (leaves/stems) with an ethanol–water solvent (8:2 *v*/*v*) was used at ambient temperature for several days (extract I), or 3 months (extract II). Extract III was prepared from dried leaves and stems from the use of ultrasound-assisted extraction (UAE).

The HPLC-DAD assay was applied to determine the cumulative capacity of the compounds present in the *Aegopodium podagraria* L. extracts, which are able to scavenge stable organic free radicals, namely DPPH. 

The DPPH radical scavenging activity was quantified in terms of the inhibition percentage of the free radical by plant extracts and the IC_50_ values in µL/mL. Regarding DPPH assay, IC_50_ is defined as the antioxidant concentration required to obtain 50% radical inhibition. As can be seen in Figure 7, the extracts showed time-dependent DPPH radical scavenging. 

It was observed that the DPPH radical-scavenging effect increased with time, to a certain extent, and then achieved plateau, even with the further increases in the time up to 45 min. The peak area value of the reduced form increases, while the free radical decreases with time. The optimal time for measuring the antioxidant capacity was set at 15 min. Owing to the content of many bioactive compounds with antioxidant properties such as phenolic acids, flavonoids, vitamins (C, E), essential oils, and polyacetylenes, obtained ethanol extracts can be categorized as fast kinetics. The representative chromatogram is presented in Figure 8. 

To compare antioxidants properties of extracts, their concentrations responsible for 50% radical-scavenging activity against 350 µM of the DPPH radical were measured by the established method under conditions allowing the reaction to run to completion (15 min), simultaneously ensuring the best detection sensitivity (517 nm). The IC_50_ values of scavenging DPPH radicals for extracts were in the range from 64.74 to 336.81 ± 0.61 µL/mL. The IC_50_ values obtained from ethanol extract prepared from fresh plants were significantly different from the IC_50_ values obtained for the same extract after 3 months. The study revealed that ethanol extract obtained from dry plants in ultrasonic bath have the highest free radical-scavenging or inhibiting activities (Table 4). The radical-scavenging activities can be also expressed as the ascorbic acid equivalent as a reference chemical. 

## 3. Materials and Methods

### 3.1. Chemicals

1,1-Diphenyl-2-picrylhydrazyl (DPPH•), ascorbic acid, and reduced glutathione (GSH) were purchased from Sigma-Aldrich (St. Louis, MO, USA). Methanol was obtained from E.Merck (Darmstadt, Germany). Water purified by an ULTRAPURE Millipore Direct-Q 3UV-R (Merck, Darmstadt, Germany) of the resistivity 18.2 MΩ cm was used to prepare all the aqueous solutions. 

### 3.2. Collection of the Plant Material and Sample Preparation

The aerial parts of *A. podagraria* plants were collected in Poland in May 2020. For extraction, the samples of 2.5 g of the fresh (extract I, II) or air-dried at room temperature (extract III) for three months (both leaves and stems) were weighted, and 100 mL of 80% (*v*/*v*) ethanol was added. The samples were kept at room temperature and darkness. Extract I was kept for 3 days, whereas extract II was kept for three months. Extract III was kept for 60 min in an ultrasonic bath. The extract was filtered through a 0.45 μm filter. The obtained samples were refrigerated at 4 °C for further investigations.

### 3.3. Determination of DPPH Radical Scavenging Activity

The DPPH radical scavenging activity of reference compounds and extracts was determined by mixing of the samples with methanolic solution of DPPH (700 µM) in separate 2 mL, amber glass vials, and measured after incubation in the dark at room temperature (20 °C). The stock solutions of the standards at concentration of DPPH-R (1 mM), ascorbic acid (5 mM), and GSH (5 mM) were prepared in methanol for the first compound or in 80% methanol for the second and third compound, respectively. The stock solutions were stored in darkness at 4 °C in glass volumetric flasks. The working standard solutions were prepared daily by diluting the stock solutions by the use of appropriate organic solvent. The radical scavenging activity of the DPPH radical by the antioxidant (AO) samples was evaluated using the following equations: % absolute decrease of DPPH-R peak area = {(*AC* (0) − *AA (t*))/*AC* (0)} × 100(4)
% absolute increase of DPPH-H peak area = {(*AC* (0) − *AA (t*))/*AC* (0)} × 100(5)
where *AC* (0) = peak area before the antioxidant addition, *AA (t*) = peak area after antioxidant addition at time.

### 3.4. HPLC Measurement

The retention factors were measured with a liquid chromatograph and an Elite LaChrom HPLC Merck-Hitachi (Merck, Darmstadt, Germany) with a DAD (Diode Array Detector L-2455) detector, pomp L-2130, and a manual sample injection valve equipped with a 20 µL loop and EZChrom Elite software (Merck, Darmstadt, Germany) system manager as a data processor. The column was a Zorbax Eclipse XDB-C18 (Agilent Technologies, Santa Clara, CA, USA); (150 mm × 4.6 mm I.D., 5 µm). The injection of blank mobile phase volumes produced visible detector fluctuations that were used as the hold-up volume. The column was thermostated at 20.0 °C ± 0.1 using a column thermostat Jetstream 2 Plus (100375, Knauer, Berlin, Germany). The elution was carried out in the isocratic mode by the mobile phase consisting of 80% methanol in water. The mobile phase was filtered through a Nylon 66 membrane filter (0.45 μm) Whatman (Maidstone, Kent, England) by the use of a filtration apparatus.

The retention data were recorded at a flow rate of 1 mL min^−1^ with online degassing using L-7612 solvent degasser at a wavelength chosen accordingly with the recorded spectra in the range of 200–600 nm. Typical injection volumes were 20 µL. HPLC measurements were performed in triplicate.

### 3.5. Statistical Analysis

In all the experiments, three samples were analyzed, and all the assays were carried out at least in triplicate. The results were expressed as the mean of the values obtained for the replications. The statistical analysis was performed using Microsoft Excel 2007 and the PQStat software. For percent of inhibition, DPPH was used one-way analysis of variance (ANOVA). Statistical significance was established at *p* < 0.05.

## 4. Conclusions

HPLC systems have been already reported as alternative techniques to spectrophotometry to measure the radical scavenging properties of various antioxidants (AOs). Stable 1,1-diphenyl-2-picrylhydrazyl radical as well as its reduced form utilized in this assay exhibit absorption within the UV-Vis wavelengths and can be well separated by the use of reversed-phase systems. Owing to this method, the interference coming from colored samples can be avoided, quantifying a well-established redox DPPH-R/DPPH-H couple. The most important requirements to apply DPPH-RP-HPLC in the aim to evaluate the antioxidant capacity are well-separated peaks of DPPH-R/DPPH-H from sample components and a proper concentration range of AO as well as DPPH radical. It should be emphasized that the LOD/LOQ values for DPPH-R/DPPH-H are even smaller in the presence of AO and achieve the level of 9.54/28.93 and 19.18/58.13 µM at 517 nm respectively, in comparison to the limits determined for the DPPH-R radical standard (35.18/112.650 µM). Attempts to detect both DPPH forms in the ultraviolet range or to measure the growth of the reduced form instead of DPPH free radical inhibition were failed. It turns out that the visible light range is the most advantageous in terms of the sensitivity of the method and the range of linearity. 

In this study, the antiradical activity tests were performed by RP-HPLC-DAD using the 1,1-diphenyl-2-picrylhydrazyl radical on common antioxidant standards ascorbic acid and reduced glutathione and different extracts (ethanol–water 8:2 *v*/*v*) prepared from fresh and dry *A. podagraria*. It turned out that the use of dried plant material as well as the reduction of the extraction time thanks to ultrasound allows obtaining extracts with the highest antioxidant potential. The IC_50_ value calculated for an extract from the dry plant prepared in the ultrasonic bath was 64.74 µL/mL. The obtained data can be used on the industrial extractive scale-up in order to optimize the production process of *A. podagraria* extracts as an efficient source of natural antioxidant agents. 

## Figures and Tables

**Figure 1 molecules-25-06005-f001:**
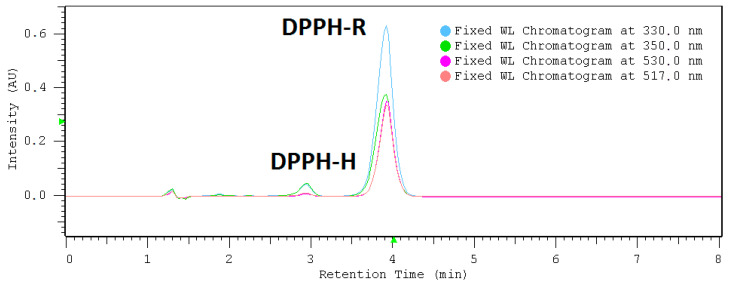
Overlapped chromatograms of DPPH (2,2-diphenyl-1-picrylhydrazyl radical (DPPH• (DPPH-R)) standard solution in methanol (0.7 mM) recorded at different wavelengths on Zorbax Eclipse column and 80% (*v*/*v*) MeOH/water.

**Figure 2 molecules-25-06005-f002:**
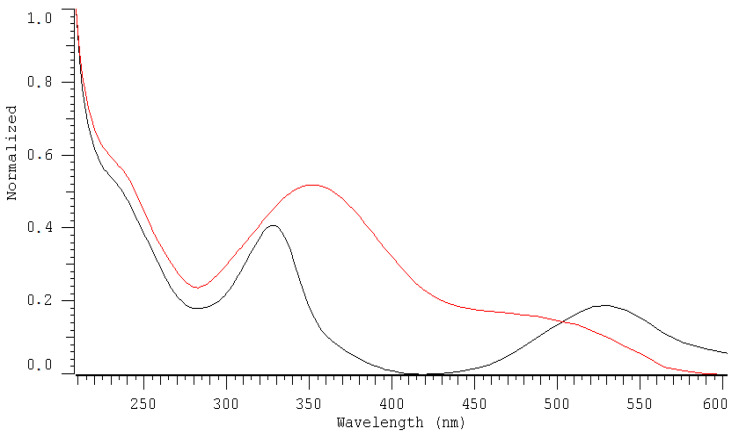
Recorded spectra in the range of 200–600 nm for DPPH• (DPPH-R-black line) and DPPH-H (red line) at 3.98 min and 2.97 min, respectively.

**Figure 3 molecules-25-06005-f003:**
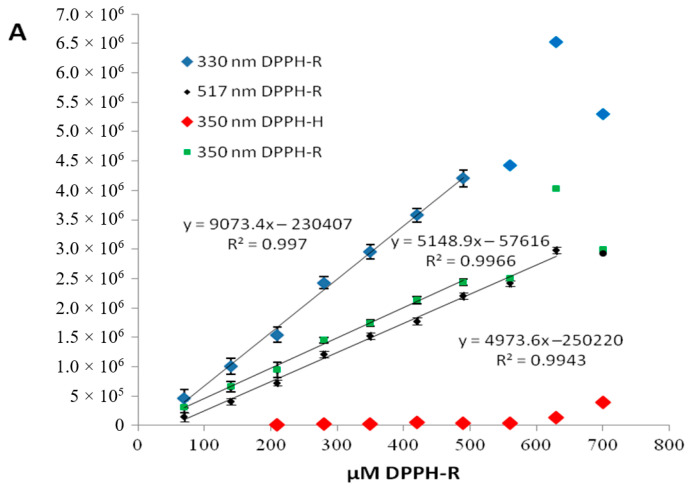
Dependence of the peak area of the DPPH radical on its concentration measured at different wavelength. Chromatographic conditions: column—a Zorbax Eclipse XDB-C18 (4.6 × 150 mm, 5 μm), mobile phase—methanol/water (80:30, *v*/*v*), flow rate—1 mL/min. The presence of the trace content of DPPH-H (2,2-diphenyl-1-picrylhydrazine) at each measured sample is illustrated in red.

**Figure 4 molecules-25-06005-f004:**
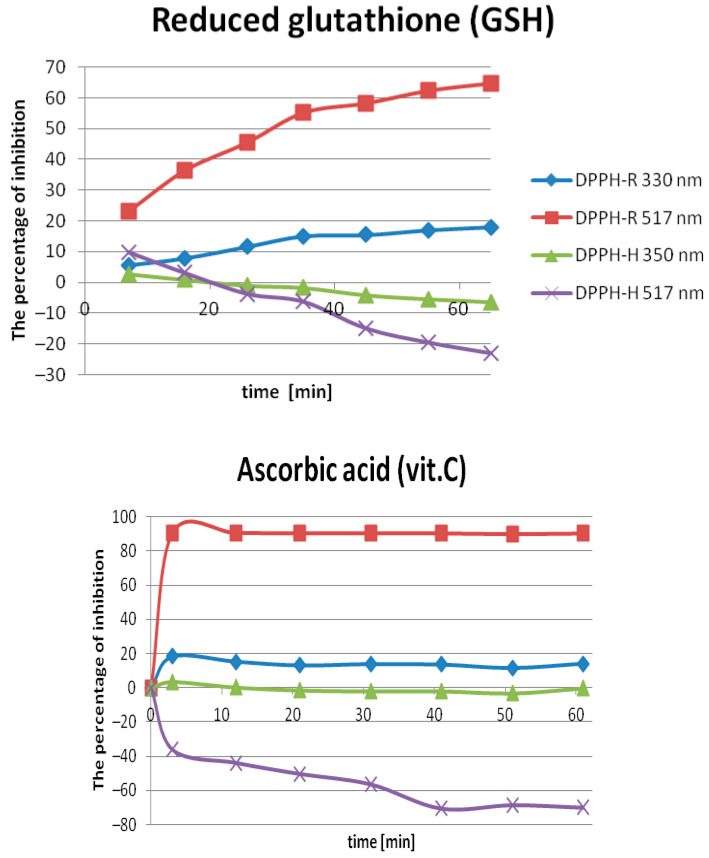
Scavenging effect of glutathione reduced (GSH) and ascorbic acid (vit. C) on DPPH free radical. Each value expressed as a mean ± SD (*n* = 3). Conditions: 500 µL of 0.7 mM DPPH in methanol was mixed with 500 µL 0.7 mM GSH, or 500 µL 0.7 mM ascorbic acid dissolved in 80% (*v*/*v*) MeOH/water.

**Figure 5 molecules-25-06005-f005:**
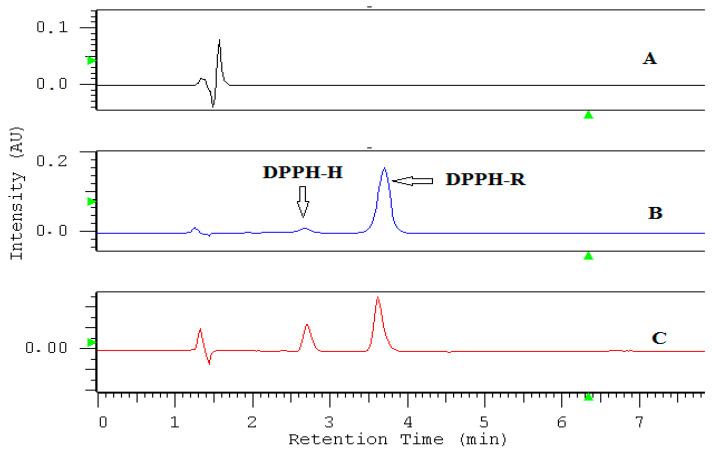
Chromatograms of 0.5 mM of ascorbic acid in 80% (*v*/*v*) methanol (**A**), 0.35 mM DPPH in methanol (**B**), DPPH reduced with ascorbic acid recorded at 517 nm after 15 min in the dark place (**C**).

**Figure 6 molecules-25-06005-f006:**
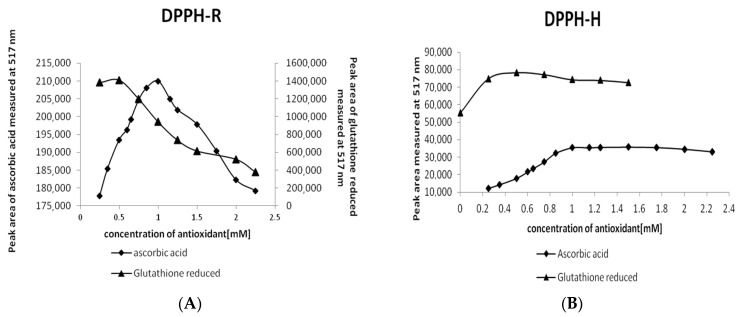
The surface areas of the DPPH-R (**A**) and DPPH-H (**B**) after reaction 1 mL of 1 mM DPPH with 1 mL of different concentrations of ascorbic acid and reduced glutathione dissolved in 80% (*v*/*v*) methanol measured by RP-HPLC-UV at 517 nm.

**Figure 7 molecules-25-06005-f007:**
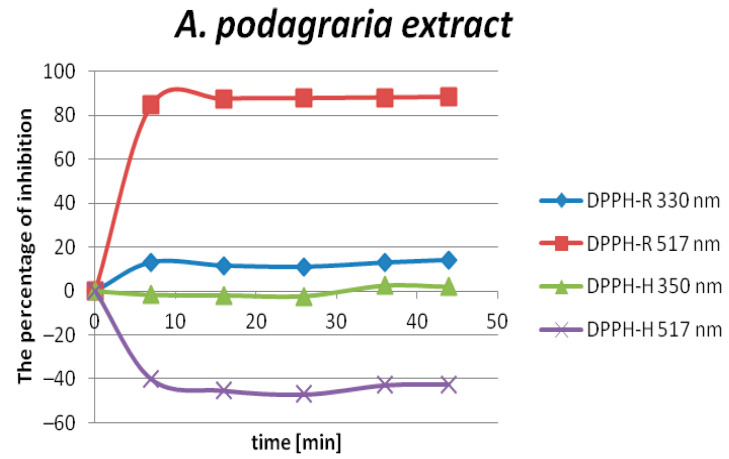
The inhibition percentage of the DPPH-R/DPPH-H couple by *A. podagraria* extracts vs. time measured at different wavelengths. Conditions: 1 mL DPPH (0.7 mM) in MeOH + 500 µL extract (I) + 500 µL MeOH.

**Figure 8 molecules-25-06005-f008:**
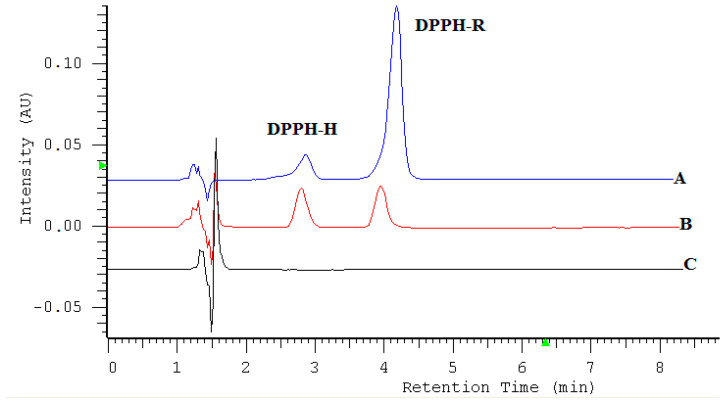
Chromatograms obtained by RP-HPLC-UV at 517 nm. (**A**) 1 mL 0.7 mM DPPH-R + 1 mL methanol; (**B**) 1 mL DPPH-R (0.7 mM) + 500 µL extract (III) + 500 µL methanol after 15 min in the dark place; (**C**) *A. podagraria* extract III 500 µL + 1.5 mL methanol.

**Table 1 molecules-25-06005-t001:** Linearity range, limits of quantification (LOQ) and detection (LOD) parameters for the detection conditions of DPPH-R/DPPH-H.

DAD nm	LOD µmol L^−1^	LOQ µmol L^−1^	Linearity Range: µmol L^−1^	Linearity Equation: (y = ax + b) *				
				a ± SD	b ± SD	R^2^	F	s_e_
The detection conditions of DPPH-R in MeOH								
330 350 517	25.6 26.9 37.2	77.4 81.5 112.7	70–490 70–490 70–630	9073.44 ± 224.42 5148.86 ± 134.11 4973.57 ± 142.24	−230,407.43 ± 70,255.84 −57,615.57 ± 41,983.95 −250,219.75 ± 56,030.77	0.997 0.997 0.994	1634.58 1473.95 1222.59	83,127.83 49,676.07 77,126.04
The detection conditions of DPPH-R in presence of AO **								
330 517	6.4 9.5	19.3 28.9	3.5–315 17.5–315	5,700,012.9 ± 69,506.8 301,066.5 ± 4882.4	−4386.17 ± 10,996.12 −953.57 ± 871.17	0.992 0.998	6725.07 3802.36	27,621.50 1630.22
The detection conditions of DPPH-H in presence of AO **								
350 517	6.4 19.2	19.5 58.1	7–315 28–315	501,769.22 ± 5957.58 122,833.91 ± 3820.14	88.77 ± 978.05 −926.77 ± 714.59	0.999 0.992	2260.64 1033.89	7093.61 1164.66

* The REGLINP function was used to calculate the statistics for a straight line using the least-squares method. The function returns the following parameters of the best-fit linear trend (y = ax + b): coefficient of determination (R2), Fisher F statistic (F), standard error of estimate (se), point of intersection (b ± SD), slope (a ± SD) with standard error values (SD) for constants. The measured parameters are the arithmetic mean of three independent determinations. ** Measurements were performed in excess of ascorbic acid after 15 min in a dark place. Conditions: 1 mL of 5 mM ascorbic acid dissolved in 80% (*v*/*v*) MeOH was mixed with 1 mL of DPPH-R in methanol at different concentrations. Measurements were performed in excess of ascorbic acid after 15 min in a dark place.

**Table 2 molecules-25-06005-t002:** *k*-Retention parameter, *A*s-tailing factor, and *N*-theoretical plate number values for DPPH standard in the HPLC system studied.

Detection Wavelength	330 nm			350 nm			517 nm		
DPPH Conc.	*k*	*A*s	*N*(EUP)	*k*	*A*s	*N*(EUP)	*k*	*A*s	*N*(EUP)
70 µM	3.31	0.76	1411	3.29	0.8	1174	3.31	0.79	3590
140 µM	2.49	0.81	1278	2.49	0.8	1010	2.49	0.88	2014
210 µM	2.05	0.88	2429	2.05	0.86	2043	2.05	0.95	3188
280 µM	2.05	0.89	2435	2.05	0.87	2080	2.07	0.87	3062
350 µM	2.05	0.85	2493	2.05	0.84	2118	2.05	0.91	3051
420 µM	2.05	0.85	2409	2.05	0.84	1927	2.05	0.94	3220
490 µM	2.05	0.86	2450	2.05	0,85	2039	2.05	0.94	3044
560 µM	2.05	0.91	2671	2.05	0.89	2348	2.05	1.00	3111
630 µM	2.03	0.87	1710	2.03	0.85	1350	2.05	0.90	3000
700 µM	2.03	0.91	2532	2.02	0.89	2217	2.05	0.88	3051

All calculations were performed using the HSM program. The following equation is used to calculate the number of theoretical plates according to EUP standards: *N* = 5.54(RT/W_1/2_)^2^, where RT = the actual full retention time of the appropriate peak, W_1/2_ = the peak width obtained by drawing tangents to both sides of the peak and calculating the distance between the two points where the tangents meet a line that runs parallel to the baseline at half peak-height. The HSM program uses the following equation to calculate asymmetry: *A*s = 1/2(1 + B/A), where A and B are evaluated at a 5% peak height of an appropriate peak. The capacity factor (*k*) is calculated as follows: *k* = t_R_/t_0_ − 1, where t_R_ = the actual retention time of the individual peak, t_0_ = the elution time of the unretained sample (thiourea).

**Table 3 molecules-25-06005-t003:** Linear regression parameters for the detection conditions of AOs by DPPH-R quench at 517 nm. Concentration of DPPH-R was 0.5 mM.

AOs	Linearity Range mmol L^−1^	Linearity Equation: (y = ax + b)				
		a ± SD	b ± SD	R^2^	F	s_e_
Ascorbic acid Reduced glutathione	1.0–2.25 0.25–1.5	−25,006.8 ± 1153.1 −731,164.8 ± 63277.0	234,157.8 ± 1862.3 1,544,272.5 ± 61,607.2	0.9895 0.9709	470.30 133.52	1308.17 66,176.70

**Table 4 molecules-25-06005-t004:** The IC_50_ of plant extracts estimation by DPPH-HPLC-DAD at 517 nm. DAD: diode array detector.

The Antioxidant Capacity	Extract I	Extract II	Extract III
Sample conc.	100 µL/mL	100 µL/mL	100 µL/mL
DPPH-R conc.	350 µM	350 µM	350 µM
The peak inhibition *	29.69%	33.38%	77.24%
IC_50_ ** ± RSD %	336.81 ± 0.61 µL/mL	149,79 ± 0.31 µL/mL	64.74 ± 0.22 µL/mL
Equivalent conc. of ascorbic acid	0.49 mM (86.30 µg/mL)	0.55 mM (96.87 µg/mL)	1.27 mM (223.68 µg/mL)

* The scavenging activity was estimated based on the percentage of DPPH radical scavenged using the following equation: The peak inhibition (%) = (Control peak area–Sample peak area)/(Control peak area) × 100. ** The antiradical activity was expressed as IC_50_ (µg/mL), the concentration required to cause 50% DPPH inhibition under defined conditions.

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
