# Peer review of "The [DPPH●/DPPH-H]-HPLC-DAD Method on Tracking the Antioxidant Activity of Pure Antioxidants and Goutweed (Aegopodium podagraria L.) Hydroalcoholic Extracts"

_molecules, 2020, doi:10.3390/molecules25246005_

Round 1

Reviewer 1 Report

This new version of the manuscript (Molecules-1044459) substantially improves the main purpose of the study (DPPH-HPLC-DAD standardization), detected in its first version (Molecules-1013544). All changes made to it enhanced the scientific contribution of the study but there are some aspects that require attention:

1) Suggested title: The [DPPH * / DPPH-H] -HPLC-DAD method on tracking the antioxidant activity of pure antioxidants and Goutweed (A. podagraria L) hydroalcoholic extracts.
2) Introduction: Too long. Suggestion: Paragraph 119 to 131 could be placed at the beginning of section 3.3., highlighting the importance of evaluating the antioxidant capacity of different fractions of Goutweed, since their specific antioxidant bioactive profile may differ.
3) Section 3.3. Should be discussed more in-depth as to relevant molecules found in all fractions, which are responsible for the observed fraction-specific antioxidant activity. Authors should also include inductive statements as to the pharmacological applications and relevant information on the industrial extractive scale-up.

Author Response

Reviewer 1

This new version of the manuscript (Molecules-1044459) substantially improves the main purpose of the study (DPPH-HPLC-DAD standardization), detected in its first version (Molecules-1013544). All changes made to it enhanced the scientific contribution of the study but there are some aspects that require attention:

Thank You very much. We appreciate the Reviewer's opinion about our efforts to improve our paper and answer all questions and doubts.

1) Suggested title: The [DPPH * / DPPH-H] -HPLC-DAD method on tracking the antioxidant activity of pure antioxidants and Goutweed (A. podagraria L) hydroalcoholic extracts.

Yes. We agree with the Reviewer and the title was improved.

2) Introduction: Too long. Suggestion: Paragraph 119 to 131 could be placed at the beginning of section 3.3., highlighting the importance of evaluating the antioxidant capacity of different fractions of Goutweed, since their specific antioxidant bioactive profile may differ.

Yes. We transferred this part to subsection 3.3.

3) Section 3.3. Should be discussed more in-depth as to relevant molecules found in all fractions, which are responsible for the observed fraction-specific antioxidant activity. Authors should also include inductive statements as to the pharmacological applications and relevant information on the industrial extractive scale-up.

We added more information and two additional references in 3.3 section. Moreover, we added a statement in the conclusion part. All changes were marked in red color in a new version of the manuscript.

Added to section 3.3

As evidenced by previous studies, the extraction conditions determine the chemical composition of extracts [48,59] which, therefore, may show different pharmacological activity and may differ in the total antioxidant potential.

It is known that goutweed is a rich source of many biologically active substances such as polyacetylenes, terpenes, coumarins, polyphenolic compounds, micro and macro elements, and vitamins (ascorbic acid, α-tocopherol). It was found that the aerial parts of this plant have a significant lipophilic antioxidant content (146.07 m /100 g - 1 fresh weight) similar to spinach, and carotenes slightly lower than in nettle [48, 60].

The evaluation of the antioxidant capacity of various goutweed extracts and the creation of their specific, bioactive antioxidant profile can help to select the most favorable conditions ensuring the extraction of natural antioxidants with pro-health effects.

To prepare the extracts, maceration of ground fresh plant material (leaves/stems) with an ethanol-water solvent (8:2 v/v) was used at ambient temperature for several days (extract I), or 3 months (extract II). Extract III was prepared from dried leaves and stems from the use of Ultrasound-Assisted Extraction (UAE).

  1. Pisoschi,A.M.; Pop, A.; Cimpeanu, C.; Predoi, G. Antioxidant Capacity Determination in Plants and Plant-Derived Products: A Review. Oxid. Med. Cell. Longev. 2016, 2016, 9130976. doi: 10.1155/2016/9130976. 
  2. Šircelj, H.; Mikulic-Petkovsek, M.; Veberic, R.; Hudina, M.; Slatnar, A. Lipophilic antioxidants in edible weeds from agricultural areas. Turk. J. Agric. For. 2018, 42, 1–10. doi: 10.3906/tar-1707-25.

Added to Conclusion:

It turned out that the use of dried plant material as well as the reduction of the extraction time thanks to ultrasound allows obtaining extracts with the highest antioxidant potential. The IC50 value calculated for an extract from the dry plant prepared in the ultrasonic bath was 64.74 µL/mL. The obtained data can be used on the industrial extractive scale-up in order to optimize the production process of A. podagraria extracts as an efficient source of natural antioxidant agents.

Once again the Authors would like to express great thanks to the Reviewer for all the substantial requirements and suggestions which allowed us to improve our paper.

Reviewer 2 Report

The authors have improved considerably the ms according to the suggestions by the reviewers and it should be accepted for publication.

Author Response

Reviewer 2

The authors have improved considerably the ms according to the suggestions by the reviewers and it should be accepted for publication.

The authors would like to express their great thanks for all the Reviewer’s suggestions which helped us to improve our work.

This manuscript is a resubmission of an earlier submission. The following is a list of the peer review reports and author responses from that submission.

Round 1

Reviewer 1 Report

Molecules-1013544

The submitted ms proposes the evaluation of total antioxidant capacity by using the DPPH method together with RP-HPLC-DAD technique which is very useful for coloured sample solutions. The procedure was applied to alcoholic extracts of  Aegopodium podagraria L.

The scientific part of the ms is interesting but requires further elaboration before acceptance for publication. The following major revisions should be elaborated:

  1. line 116: 80% ethanol should be corrected to 80% (v/v) ethanol. The same throughout the ms.
  2. line 128: The term AO is confusing. Is it AC (antioxidant capacity) as written on line 132? The same on line 247.
  3. line 141: It should be 20.0 oC±0.1
  4. Figure 2: Since the time intervals 2.97 and 3.98 min are given, it seems that this is after mixing of DPPH with antioxidant. Please give amounts. Also it is not clear what is the red line and the black line of the figure.
  5. Figure 3 and Table 1: Figure 3 can be removed since all parameters are presented in Table 1. All significant figures should be corrected. For example, LOD equal to 25.5519 μmol/L is impossible. 25.6 seems more reasonable. Correlation coefficients should be presented with all 9 plus the first not 9. Example 0.9966 should be 0.997. All correlation coefficients should be given together with the number of measurements otherwise there is no statistical meaning. How was the SD of a and b calculated?
  6. Figure 4: The authors present linear regression equations but the lines are not linear. Figure 4 should be removed and results should be presented in a Table.
  7. Table 3: Please correct correlation coefficients and give number of measurements. How was SD for a and b calculated?

Reviewer 2 Report

The paper describes a useful combination of HPLC and DPPH antioxidant assay for the determination of the antioxidant properties of A. podagraria L.. The authors concluded that the use of HPCL provides advantages over spectrophotometry and documented the strong antioxidant effects of the studied plant. The paper is well written, easy readable and contains valuable information. The study was well designed, with logical phases which led to reliable results. The conclusions are supported by the recorded experimental data. It is therefore my opinion that the article can be published in its current form.

Reviewer 3 Report

The authors evaluated the radical scavenging capacity of Aegopodium podagraria L. extracts [leaves/stems; fresh (ext I and II) and air-dried (ext III) samples], ascorbic acid (5mM), and reduced glutathione (5 mM) by a standardized [DPPH*/DPPH-H]-HPLC-DAD (517 nm) method. The authors report in detail many aspects related to the standardization of the chromatographic method, leaving aside the description of the antioxidant capacity of the sample under study.

  • The manuscript is written in such a way that more than 70-80% of it refers to the standardization details of the chromatographic method (DPPH-HPLC) and only 20-30% refers to the study sample (Aegopodium podagraria).
  • The DPPH-HPLC-DAD method used here was initially developed by Blois (DOI: 10.1038/1811199a0), described by Yamaguchi et al (Ref. 49), and later standardized by Boudier (Ref. 50). These and other authors (DOI: 10.1021 / jf500180u, 10.1016/j.foodchem.2011.07.127) have proved that the DPPH * -DPPH-H reaction depends on many factors such as the type of extraction solvent, temperature, pH, and analyte concentration, testing the same controls used in this study. Moreover, this body of evidence points out the fact that this chromatographic method provides almost the same results that the spectrophotometric method at 500 (blue)-520 (cyan) nm. Nevertheless, the method has been used (with no further explanation as to the experimental conditions used) to evaluate the antioxidant capacity of natural products (e.g. DOI: 0.1371/journal.pone.0170141, 10.1111/1750-3841.13730) whose extracts may have molecules absorbing in the 500-520 region.
  • As for the antioxidant capacity of fresh (extracts I and II) and air-dried (extract III) Aegopodium podagraria (“Goutweed”), it has been reported by other authors (). It is noteworthy that alcoholic extracts from this plant contain a wide range of volatiles (doi:10.1016/j.proche.2009.12.022, 1016/j.proche.2009.12.022) and phenolic compounds (DOI: 10.22616/foodbalt.2017.028; ) but ascorbic acid or glutathione have not been reported. The effect of extractive conditions and solvents on the antioxidant capacity of leaves and stems of this plant has been reported (10.2478/pjct-2019-0024).

Based on the above, the novelty of this study is low, at least in its current form. To increase its scientific soundness, the authors are invited to consider the following:

  • English grammar and syntax should be improved. The manuscript readiness will improve if it is sent to a formal translation agency.
  • Title. It does not reflect is content (DPPH-HPLC-DAD standardization > Goutweed antioxidant capacity).     
  • Abstract. It should be more concise without sacrificing important results, expressed in a more quantitative way, including statistical differences (p-values).
  • Introduction. The whole section should be reconstructed in such a way that it reflects its current content/experimental design (DPPH-HPLC-DAD standardization > Goutweed antioxidant capacity)
  • Tables and figures. Many of them require footnotes or statistical differences.
  • Results and discussion. Be consistent with all abbreviations troughout the manuscript and include their meaning the first time they are mentioned. Avoid any inductive statement not related to the presented data.
  • Methods. Ascorbic acid and glutathione are not good antioxidant references to the antioxidant composition of the plant material.
  • References. Please check the reference format according to authors’ guidelines, particularly DOI, WEB links, and upper-case titles. Include relevant studies dealing with the DPPH-HPLC method and the scientific basis of SET/HAT mechanisms of DPPH under different assay conditions.